# Acceptability and feasibility of weight management programmes for adults with severe obesity: a qualitative systematic review

Zoë C Skea,[1] Magaly Aceves-Martins,[1] Clare Robertson,[1] M De Bruin,[2] Alison Avenell,[1] on behalf of the REBALANCE team

[1]Health Services Research Unit, University of Aberdeen, Aberdeen, UK
[2]Health Psychology Department, University of Aberdeen, Aberdeen, UK

**Correspondence to**
Dr Zoë C Skea;
z.skea@abdn.ac.uk

## ABSTRACT

**Objectives** To improve our understanding of the acceptability of behavioural weight management programmes (WMPs) for adults with severe obesity.

**Design** A systematic review of qualitative evidence.

**Data sources** Medline, Embase, PsycINFO, CINAHL, SCI, SSCI and CAB abstracts were searched from 1964 to May 2017.

**Eligibility criteria** Papers that contained qualitative data from adults with body mass index (BMI) ≥35 kg/m² (and/or the views of providers involved in their care) and considered issues about weight management.

**Data extraction and synthesis** Two reviewers read and systematically extracted data from the included papers which were compared, and contrasted according to emerging issues and themes. Papers were appraised for methodological rigour and theoretical relevance using Toye's proposed criteria for quality in relation to meta-ethnography.

**Results** 33 papers met our inclusion criteria from seven countries published 2007–2017. Findings were presented from a total of 644 participants and 153 programme providers. Participants described being attracted to programmes that were perceived to be novel or exciting, as well as being endorsed by their healthcare provider. The sense of belonging to a group who shared similar issues, and who had similar physiques and personalities, was particularly important and seemed to foster a strong group identity and related accountability. Group-based activities were enjoyed by many and participants preferred WMPs with more intensive support. However, some described struggling with physical activities (due to a range of physical comorbidities) and not everyone enjoyed group interaction with others (sometimes due to various mental health comorbidities). Although the mean BMI reported across the papers ranged from 36.8 to 44.7 kg/m², no quotes from participants in any of the included papers were linked to specific detail regarding BMI status.

**Conclusions** Although group-based interventions were favoured, people with severe obesity might be especially vulnerable to physical and mental comorbidities which could inhibit engagement with certain intervention components.

## Strengths and limitations of this study

► This is the first review of key findings from qualitative studies exploring views of weight management programmes for adults with severe obesity (body mass index (BMI) ≥35 kg/m²).

► Qualitative studies have a key role to play in understanding how factors facilitate or hinder the effectiveness of interventions, and how the process of interventions are perceived and implemented by users.

► Across the 33 papers, specific participant characteristics were inconsistently and poorly reported (if at all).

► Although the mean BMI reported across the papers ranged from 36.8 to 44.7 kg/m², no quotes from participants in any of the included papers were linked to specific detail regarding BMI status.

## INTRODUCTION

There has been a continued increase in body mass index (BMI) ≥35 kg/m² (denoted here by the term 'severe obesity') in adults in the UK.[1 2] As BMI increases, obesity-related comorbidities, social, psychological and economic consequences increase, with the potential need for greater support for help with weight loss. In the UK, having severe obesity, with or without comorbidities, may be a referral criterion for Tier 3 specialist weight management services in the obesity pathway, prior to Tier 4 services for bariatric surgery.[3 4] Effective weight-loss services may reduce the need for bariatric surgery, and could also increase the effectiveness of subsequent bariatric surgery.[5] Current National Institute For Excellence (NICE) and Scottish Intercollegiate Guidelines Network (SIGN) guidance on weight management for obesity does not distinguish between obesity (BMI 30 to <35 kg/m²) and severe obesity (BMI ≥35 kg/m²); and public health guidance

excludes evidence on weight-loss programmes for obese people with comorbidities in the UK.[3 6 7] This implies that Tier 3 services are being created and money is being spent without an appropriate systematic review that clarifies what works for people with severe obesity (and their comorbidities).

Qualitative studies have a key role to play in understanding how factors facilitate or hinder the effectiveness of interventions, and how the process of interventions are perceived and implemented by participants. This qualitative systematic review was conducted as part of a larger systematic review funded by the UK's National Institute for Health Research Health Technology Assessment Programme[8] and aimed to improve our understanding of the feasibility and acceptability of non-surgical weight management programmes (WMPs) for adults with severe obesity and programme providers. Previous qualitative reviews have been undertaken[9 10] but these have not focused on WMPs that are designed for or include people with severe obesity.

Our broad initial research questions included 'What is it like to engage with (or be a provider of) weight-loss interventions for adults with severe obesity?' and 'What is it about interventions for adults with severe obesity that makes them helpful or unhelpful?' Our review also considered issues around what might motivate people to decide to engage in such programmes.

This paper focuses on the main themes that emerged from the qualitative review of included studies. These themes shed light on (1) motivating factors for engagement; (2) components of WMPs participants described valuing; and (3) general challenges for engagement.

## METHODS
### Searching and identification of relevant studies
A systematic search was conducted in June 2016 and updated during April/May 2017 for published papers that contained qualitative data from adults with BMI $\geq 35 \, \text{kg/m}^2$ (and/or the views of providers involved in their care) and considered issues relating to weight management (see online S1 Appendix for search strategies and S1 ENTREQ Checklist). Two researchers (ZCS and MA-M) independently screened titles, abstracts and selected full text papers. Where consensus could not be reached regarding eligibility, a discussion at a research team meeting took place.

We included studies that fitted into the following broad categories:
1. Qualitative and mixed-methods studies linked to eligible randomised controlled trials (RCTs) (from our other review), including any qualitative data reported as part of papers reporting quantitative outcomes.
2. Qualitative and mixed-methods studies linked to ineligible RCTs and identified non-randomised intervention studies including any reported qualitative data.
3. Qualitative studies not linked to specific interventions that drew on the experiences and perceptions of adults

with BMI $\geq 35 \, \text{kg/m}^2$ (and/or providers involved in their care) providing they reported data specifically relating to views/experiences of strategies for weight loss.

### Analysis and synthesis
There are several approaches that can be used for synthesising the findings of qualitative studies.[11 12] While being aware of the differing philosophical stances underlying various approaches to qualitative synthesis, we chose to adopt a pragmatic approach to our work in this area, which specifically aims to synthesise data that are relevant to informing policy and practice.[10] Our pragmatic approach drew on a 'realist' perspective[12 13] as we were concerned with trying to find out not only 'what works' for weight management for this group of adults and intervention providers, but also 'for whom, and under what circumstances'. At the same time, our approach was informed by and used aspects of review methods such as thematic synthesis[14 15] and analytical approaches developed from methods of inquiry such as grounded theory.[15]

In order to collate and synthesise the available primary research, two authors (ZS, MA-M) each read and systematically extracted data from the included papers, shared notes and discussed study findings and interpretations during a series of group meetings. The papers were initially organised according to the categories described above but, as inductive analysis progressed, papers were grouped, compared and contrasted according to emerging issues and themes. We used a data extraction form, which summarised the main findings and original authors' discussion points and to note our own critical and interpretive comments on the papers. We then used these to facilitate the process of comparing and contrasting themes both within and across papers in order to develop cumulative insights into the mechanisms that are likely to impact on decisions to join and decisions to stay in or drop out of WMPs.

### Study quality
The retrieved publications were appraised for methodological rigour and theoretical relevance independently by two reviewers using Toye's recently proposed criteria for quality in relation to meta-ethnography.[16] They suggest two core facets of quality for inclusion in syntheses of qualitative evidence, namely (1) conceptual clarity: how clearly has the author articulated a concept that facilitates theoretical insight; (2) interpretive rigour: what is the context of the interpretation; how inductive are the findings; has the interpretation been challenged? Two reviewers made notes regarding quality and results were compared and discussed.

### Patient and public involvement
The REBALANCE Advisory Group included a mix of professional and lay members identified through team contacts (a clinician; dietician; policymaker; and three lay people who had all experience of severe obesity and use of

related services) who offered advice throughout various stages of this project including during initial discussions around the choice of appropriate research questions to attempt to answer and areas of interest for this review, and our other suite of reviews which considered issues around intervention effectiveness and cost-effectiveness.[8] Results were disseminated at a final project meeting in 2018 at which the advisory group were present.

## FINDINGS
### Description of studies
The database search produced 4710 abstracts (see online S1 figure for the PRISMA diagram providing information on the flow of studies through the review). Four additional papers were identified from included RCTs. In all, 33 papers met our inclusion criteria.[17–49]

The focus and key study characteristics of the 33 papers are outlined in online supplementary S1 table. The identified papers reported research conducted in seven countries (USA n=12; UK n=11; Norway n=3; Spain n=1; Canada n=2; Australia n=3; Mexico n=1), and published between 2007 and 2017. Seven papers were linked to broader intervention studies:[20 21 23 30 42–44] Seven papers were classed as category A; 24 category B; and 2 category C. As can be seen from online supplementary S1 table, the studies had varying aims, but all offered insights into stakeholders' perceptions of weight-loss strategies and programmes.

Although all the included papers provided some qualitative data for analysis, five of these provided qualitative data in the form of responses to open-ended survey questions within structured questionnaires.[22 32 37 46 49] Of those studies that used qualitative methods to collect their data, findings were presented from a total of 644 participants and 153 programme providers (mostly from interviews or focus group sessions).

Across the 33 papers, specific participant characteristics were inconsistently and poorly reported (if at all). Only 16 out of 33 papers provided any details. Information on sex was provided for 588 participants (out of 644 of those who specifically took part in qualitative evaluations)—372 female; 216 male. Age was reported across 15 papers, with the range being 19–88 years. Six of these papers provided mean age with the range being 40.2–67 years. BMI for those involved in qualitative evaluations was reported in nine papers. Of those that provided a mean, this ranged from 36.8 to 44.7 kg/m$^2$. Only four papers gave details of participants' ethnicity; from 188 participants, 35 were reported as being from ethnic or racial minorities. Furthermore, 14 papers specifically stated that study participants had a range of additional physical and/or serious mental health problems (eg, osteoarthritis, chronic pain, schizophrenia, post-traumatic stress disorder). It was also apparent across other included papers from quotes and/or author comments that many participants had a range of similar comorbidities.

Although no included papers provided qualitative data from those who had been invited to join a programme, but had declined to take part at recruitment stage, some papers reported including participants who had not fully engaged with programme activities (being described as 'low users'; 'quitters' or 'drop outs').[17 24 25 36]

The WMPs varied in the types and formats of support offered. Some programmes involved predominantly face to face interaction and activities with other participants and/or programme staff.[24 27 29 31–35 40 45 47] Two involved more remote forms of support (eg, email, telephone, text contact).[41 46] Other studies included and evaluated a mix of formats that also varied in intensity.[17 19 23 25 30 36 37 42–44 48 49]

Programmes incorporated a variety of tools and theories designed to support behaviour change and to help people lose weight. For example, tools such as diet diaries;[24 37] workbooks;[42–44] pedometers;[36 37 48] food logs;[17 47] conversation maps;[22] interactive monitoring devices;[46] social media group interaction;[19] daily text messages;[41] buddying.[37] They also included a range of behaviour change theories (BCTs) and/or psychological support.[20 21 26] For example, goal setting;[32 33 36] motivational interviewing;[33] mindfulness;[35] self-determination theory-based support;[24] regulatory focus theory;[41] self-regulation and cognitive behavioural techniques.[17 23 27 30 31 33 36 42–44] Readiness to change and self-monitoring and feedback was also included[47] along with psychotherapeutic sessions;[34] emotional freedom therapy;[33] neurolinguistic programming;[33] solution focused therapy;[33] social learning theories.[40]

### Findings from the review—participants
This section of the paper discusses the views of participants who chose to engage with WMPs. It considers motivating factors for their initial engagement; components of the WMPs that they described valuing; and then outlines more critical reflections and challenges for engagement (see online supplementary S1 Conceptual diagram for an illustrative representation of key issues). The subsequent section of the paper discusses similar issues from the perspective of WMP providers.

### Motivating factors for engagement in WMPs
Several papers provided insights into what had motivated prospective participants to take part in a specific WMP.[24 26 27 31 33 35 47] Important 'push' factors were sometimes personal to participants. For example, expressing a desire to do something about their weight/poor physical fitness for themselves (eg, as a result of growing health concerns and/or recent personal health scares) and also feelings of accountability to their families (eg, stating that they wanted to be more engaged in activities with family members, as well as being there for family for as long as possible). Others recounted familial past experiences of health problems due to obesity or their own sudden and rapid weight gain due to mental health medication. For example:

### Recent personal health scares

I was told I was at risk of becoming diabetic (No sample characteristics provided).[33]

### Feelings of accountability to their families

I've had two kids in the last three years… that was part of the motivation… just getting fitter for my kids…I need to be aboot [about] for as long as possible (Male).[31]

### Familial past experiences of health problems due to obesity

My dad was a big guy and he developed diabetes, and he had to have surgeries and all kinds of stuff. I don't want to do that later in life (intervention arm; no other sample characteristics provided).[47]

### Sudden and rapid weight gain due to mental health medication

When I went on Zyprexa I gained a hundred pounds, very quickly. And that was really frustrating for me (control arm; no other sample characteristics provided).[47]

Some participants described motivators that were apparently related to certain aspects of the programme intervention itself. For example, because it was perceived as being endorsed as credible by health professionals; perceived as being novel and exciting in some key way, and also because it provided an opportunity to engage with the intervention in a place that was valued:[26 27 31]

When I first went in there I thought this is great. I am going to diet at my doctor's surgery. Knowing that it was at my doctor's surgery gave me a big 'oof' (no sample characteristics provided) [NB: We interpreted 'oof' as meaning that a WMP being endorsed by and delivered at the surgery gave this person a boost].[26]

Although one paper highlighted that decisions to join a WMP were sometimes difficult and that some participants had expressed initial apprehension around taking part,[31] no included studies provided data about those who were invited to join but declined to take part at recruitment stage.

### Components of lifestyle programmes participants described liking or valuing

We examined various aspects of WMPs that participants described valuing. In doing so, we were interested in the range of factors that might motivate those participants to join in the first place, and to continue to stay in the programme. We were also interested in the factors that they described as having assisted them to change aspects of their behaviour or ways of thinking. All but two papers were set within the context of a WMP. The two included papers that were not linked to a specific intervention[38 39] also provided data regarding perceptions of weight-loss strategies and engagement in diet and lifestyle programmes and were useful in this context. We found there was variation in what participants described as

valuing within their WMP, demonstrating that a one size fits all approach is unlikely to be appropriate. We noted some key recurring themes in relation to what participants valued, and we grouped these around aspects that related to (1) the overall setting or style of the programme; (2) the people (both other participants and health professionals/support staff) within the programme setting; (3) the type of interaction/support offered; (4) dietary elements; (5) physical activities; and (6) programme tools and theories designed to support behaviour change. These are discussed below.

### Overall setting or style of the programme

The overall setting of the programme was important for motivating people to decide to engage. It also seemed important for motivating them to stay in and keep going with the various intervention activities. Some participants described their programmes as being exciting or novel in that they perceived them to be different to interventions they had tried previously. For example, being focused on physical activity rather than dieting[24] or being focused on changing overall attitudes towards eating rather dieting per se.[35 43] An important consideration was the extent to which they could 'relate' to the nature of the programme (including how it was presented to them at recruitment) and how well it appeared to match with their own identities and values:[24 31 35 39]

…the main thing that drew us to it was because it's [at a football club] (Male).[31]

I always think somebody approaching you one-on-one is better. They can post all the weight loss you know pamphlets out there…I was hooked right away because somebody took the time to really explain it and take her time to do that (Female).[35]

Several participants positively contrasted their overall perceptions of the WMPs with previous negative views towards other WMPs they had engaged with. For example, WMPs which were perceived as being too 'feminine' or in some ways humiliating and embarrassing, or being perceived to be overly preoccupied with dieting:[24 25 29 32 33 39]

If you go to a slimming class you feel that you've made a fool of yourself or you get weighed and you've put on half a pound or a pound, and then you don't want to go back the next week so you don't go back (Coaching group arm; no other sample characteristics provided)[25]

Well, I think it's (WHEEL) appealed to me because I won't be dieting…I am obsessed with dieting me (Female).[24]

…spent many useless years at weight watchers with various leaders but never felt confident and in control or had the motivation I have now (No sample characteristics provided).[32]

### Importance of the people within the programme setting (for fostering a sense of accountability)

A recurring theme was the value participants placed on perceiving themselves to be part of a like-minded group of individuals—individuals who faced similar issues, and who had similar physiques and personalities.[19 22 24 25 29 31 34] For example:

> I do not feel so ashamed of my body here. We are all in the same situation, you see, which is really nice (Female).[29]

These perceptions seemed to foster a strong group identity and related 'accountability' or responsibility to other participants and programme providers. This was apparently important for people in motivating them to stick with the programmes and to not let their fellow participants down by dropping out or not sustaining behaviour changes:[17 19 24 25 31 35–37 47]

> So, you didn't want to disappoint yourself, but you didn't want to disappoint … your friends now either (No sample characteristics provided).[35]

Many participants discussed the importance of their interactions with healthcare staff within the programmes.[17 24 25 27 29 32–35 37 40 43 45 49] They seemed to value the positive, friendly and non-judgemental encouragement received. They also discussed feeling accountable to programme staff which helped with motivation. These aspects seemed to act as positive 'pulls' for staying in the intervention and helping to sustain behaviour change:

> I think I just like talking to you [programme leader]. And I suppose I feel that if I don't do it [the programme] then I'm letting you down (Female).[24]

> She is my motivator… and she makes me keep a record of my diet (Female).[29]

### Type of interaction/support offered

Although not universal, many described particularly valuing the social interactivity of group-based programme activities along with intensive support from/interaction with programme staff.[17 19 24 25 28 31 32 34–36 40 47 48] This appeared to function strongly as a motivator to maintain engagement with the WMPs by fostering feelings of accountability and by helping to ensure the achievement of preset goals:

> Oh God I haven't done what I should of done and I promised to do it and I know that isn't what's supposed to spur you on but it I think it does (Regular support group; no other sample characteristics provided).[25]

> [discussing feedback from programme staff]…great encouragement when the results are positive and a way to improve if the results are not so good (No sample characteristics provided).[32]

Participants discussed appreciating when the timing of support offered was flexible and could fit around their needs.[25 35 37] Several wanted more support than was offered within the programmes (eg, more frequent contact and for a longer duration than the programme currently allowed).[25 36 46 49] Many expressed concern about support ending postintervention[24 25 29 35 41 47] with the suggestion that diminishing intensity of programme activities and/or programme cessation could cause problems for maintaining behaviour change patterns if group interaction and support were key parts of it:

> I cannot do it without her support, it just wouldn't work (Female).[29]

Some WMPs involved predominantly face to face interaction and activities with other participants and/or programme staff.[24 27 29 31–35 40 45 47] In contrast, others involved more remote forms of support (eg, email, telephone, text contact).[41 46] Some studies included and evaluated a mix of formats that varied in intensity.[17 19 23 25 30 36 37 42–44 48 49] Many participants discussed valuing the social interactivity of the inperson group-based activities.[19 24 25 31 35 36 47] Where it was discussed and compared, participants tended to value and desire human contact over more remote forms of support.[36 46] This preference seemed to be linked to incentivising people to stay committed to the various programmes and was important for making participants feel accountable to a likeminded group of individuals.

### Dietary elements

Some WMPs provided detailed dietary advice regarding food choices, while others specifically described interventions as 'non-dietary' (nevertheless, incorporating behavioural change theories to support attitudinal changes towards food and eating patterns). Participants tended to describe valuing the flexibility and variety of diet formats.[24 35 36 40] This seemed important for helping them to 'normalise' and stabilise their eating habits, particularly as many had attempted diets over a period of many years (without success) leading them to develop negative and unhealthy relationships towards food:[24 35 36 40]

> The other programs told you not to eat this or that and you were afraid to go back if you hadn't lost weight and …they tell you that you can eat everything but you yourself have to control the amount…You make up the diet every day and that's very motivating (Female).[40]

### Physical activities

All of the WMPs incorporated some attention to increasing physical activity. While some participants described struggling to engage in exercise for a variety of reasons, many participants described the positive psychological and physical benefits they experienced from exercising:[19 24 29 33 47]

> When I first started I could hardly walk…now I can walk 300–400 yards…if this project has done nothing

else it has helped me to walk (No sample characteristics provided).[33]

When it was offered as part of the WMP, participants discussed valuing the flexibility of being able to choose from a variety of exercise formats and approaches.[24 36]

### Programme tools and BCTs designed to support behaviour change

Although not universally popular,[17 24 36 46 47] participants described the incorporation of tools (eg, food logs, goal setting, regular text messages, telemonitoring devices and conversation maps) as being motivating, and helpful for the purposes of education and learning, describing how they helped to facilitate self-awareness of and reflection on eating and other behaviour patterns:[17 22 36 37 41 46–49]

I found it to be very enlightening. It made me start to look at foods differently. It has given me a more conscious outlook on how to control my diabetes and the importance of exercise (No sample characteristics provided).[22]

What really helped me was having somebody go over the food log every day. That was the big thing (No sample characteristics provided).[17]

Participants discussed the positive psychological changes they experienced with regards to their relationship to food/body image, which seemed to relate to the BCTs employed within some of the WMPs (eg, mindfulness and self-determination theory-based support).[17 24 27 35]

### General challenges for engagement in WMPs

Despite the numerous positive comments from within the data with regard to programme engagement, participation was not straightforward for everyone who took part. General challenges resulting in decreased engagement (or success) related to a number of factors. Sometimes, these involved the timing of clinic appointments;[37] cost of travel to appointments;[33 48] general low self-efficacy;[26] family members not being on board, such that behavioural changes were difficult to sustain.[34 47] Others described factors which could be described as life getting in the way (eg, holidays, social events, bad weather as disincentive to exercise).[47]

It was apparent that participants experienced a range of comorbidities, including some serious mental health issues.[18 19 36–39 46–48] Sometimes these specific illnesses presented challenges for motivation and continuing engagement, for example, feeling too ill to focus on weight/feeling too ill to care or to be motivated:[33 36 39 40 47]

Because of the ME [myalgic encephalopathy] I'm sleeping fifteen or more hours a day, and so exercise is out of the question because I can't even walk to the end of the road (Female).[38]

### Critical reflections on specific components of WMPs
#### Type of interaction/support offered

The social interactivity of group-based programme activities was not universally valued by all, with some describing a reluctance to discuss issues within a group setting.[19 27 28 40 45 48] This was perhaps particularly pertinent in studies where participants had additional mental health issues:

I know the importance of the program is to be together, but at the beginning you don't know these people, some of us have problems interacting with people we don't know (No sample characteristics provided).[19]

It's just I don't like to be around people (No sample characteristics provided).[48]

I prefer to talk in private as I suffer from panic attacks (No sample characteristics provided).[45]

One study[44] included data that suggested some participants were guilty about using up what they perceived to be too much of their healthcare provider's time (in an intervention involving regular GP visits):

I must admit I felt frequently embarrassed that I was taking up a lot of my GP's time (No sample characteristics provided).[44]

### Dietary elements and physical activities

Although the majority of participants tended to describe valuing the flexibility and variety of the diet formats offered,[24 36 40 49] views were sometimes mixed with regard to diets, with a few wanting more prescriptive and structured eating plans than were offered. Participants often discussed appreciating when programmes apparently emphasised changing attitudes towards food and eating over promoting a specific diet per se:[24 36 40 49]

I think [having a set meal plan to follow] would have been to a certain extent easier at the beginning, but I don't think it would of actually adjusted my attitudes and thinking which it [POWeR+] has done (Male; 64 years; face-to-face support; high user).[36]

However, sometimes participants stated that their programme (or their primary care providers) tended to over emphasise diet rather than, for example, addressing issues around exercise, sleep or addiction problems:[39 47]

…there was no support counselling-wise as to why I have the issues I have with food… (Male).[39]

While many participants described the positive psychological and physical benefits they experienced from exercising,[19 24 47] others described struggling to engage in exercise. Some described disliking the perceived high intensity of the exercises (eg, feeling uncomfortable with sweating).[24 28 29] Others discussed how their various physical or mental health comorbidities could prohibit them from full engagement in activities:[18 24 28 29 36–39 47]

Exercise is the best [to lose weight] and I get all this physical therapy exercise and all of that just increases my pain, which reduces my desire to have any exercise (No sample characteristics provided).[18]

I think for me, with my disability it was difficult to engage with some of the activities recommended (No

sample characteristics provided).[37]

## Programme tools and BCTs designed to support behaviour change

Participants suggested that many of the WMPs' tools and theories were helpful to them for reflecting on their habits and behaviours and for helping them to positively change their attitudes. However, some participants described these tools as being somewhat intrusive and sometimes inflexible in nature. For example, some participants described disliking food logs and found food diaries/goal setting/daily self-weighing and the monitoring of exercise as excessive and too confrontational.[24 36 46 47] Others reported that programme staff did not appropriately monitor and feedback on progress:[17]

I mean no one ever looked at it [food diary]. No one ever asked for it. I just did all the work, like, for nothing because no one ever asked me for it (No sample characteristics provided).[17]

Others expressed frustration with the perceived inflexibility of tools designed to record behaviour and activities and to support behaviour change. For example, not being able to record life events and/or comorbidities that might help to explain lack of achievement regarding weight loss:[36 41]

I thought that might be useful [to] have something [to] explain why things are going as they are going (Female; 59 years, remote support; high user).[36]

I would want to tailor the messages [daily text messages] to the things that I was most struggling with (No sample characteristics provided).[41]

With regard to psychological support, two papers highlighted that some people wanted more counselling for non-direct weight issues, such as mental health, recognising that these additional problems had implications for weight management.[39 46] In contrast, although many participants discussed the various positive psychological changes they experienced (which seemed to relate to the BCTs/counselling employed within some of the WMPs), others found personal development classes challenging and confrontational and questioned their appropriateness:[27]

I cannot benefit from it [the personal development classes]. I will never open up in that room and talk among others (Male).[27]

## Findings from the review—provider participants

Ten of the included papers provided qualitative data from a range of WMP providers.[20 21 23 26 28 30 36 41–43] Seven of these papers were linked to one of three of the same interventions. Programme providers who provided qualitative data were described as primary care providers;[23 30] nurses;[36] GPs and consumer representatives;[43] GPs;[42 44] mental healthcare workers, dietitians and nurses;[20 21] GPs,

weight management advisors, practice nurses[26] and key personnel working at a residential weight-loss centre.[27]

## General impressions of being involved in WMPs

With the exception of one study, in which some GPs were reportedly less enthusiastic,[26] views about being involved in a WMP were generally very positive. Health professionals acknowledged that engagement was potentially very useful for them for facilitating a conversation around weight loss with participants—recognising that this can often be challenging in their everyday practices.[36 42–44]

However, the authors of one study[20] noted that discussions about weight tend to be embedded within the context of conversations about other health issues (rather than being discrete or stand-alone). They argued that this could act as a potential barrier with regards to the implementation of WMPs within primary care:

I don't have patients that come to see me just for obesity or…just one thing…yes they're one of my diabetic patients but … we're talking about their cholesterol today or their blood pressure and their weight another day (Nurse, no other sample characteristics provided).[20]

## Motivating factors for participants'/provider engagement in WMPs

One paper included some insights from the perspectives of programme providers about what motivated prospective participants to take part in a WMP.[23] Healthcare providers involved in WMP delivery described how they regarded participants' perceptions of their professional 'buy in' to the intervention study (ie, endorsement) as important and influential regarding their decisions to take part.[23] One study (linked to two papers)[23 30] reported unusual success at enrolling men which programme providers attributed to their endorsing it as a 'medical' programme:

I think that [our affiliation with a research institution] helped make it into a legitimate type of program that [our patients] would have confidence in, not just one of these wild watermelon diets or things like that (Primary Care Provider, no other sample characteristics provided).[23]

In terms of disincentives towards retention in such WMPs, some providers reported that participants could sometimes have unrealistic expectations about weight loss, not fully understanding programme goals and commitment and wanting a 'quick fix':

What they wanted was a quick fix…They want to lose pounds very quickly. And it doesn't happen… (GP, no other sample characteristics provided).[26]

Only one study[26] provided data around barriers and facilitators to health professionals' own engagement with a specific WMP. They described how clinicians' preconceived beliefs and attitudes towards integrating WMPs

into primary care settings were important and they noted that engaged practices (as opposed to less engaged practices) were characterised by active GP participation and 'buy in'.

## Importance of the people within the programme setting (for fostering a sense of accountability)

In keeping with some key findings from participants across the included papers, programme providers reflected on the importance of WMPs for creating a sense of accountability both for themselves as professionals (by increasing their responsiveness and sensitivity to their participants' weight management plan and needs) and for participants continued engagement, motivation and success:[23 42]

> …I think it just made me be more sensitive…I've been kinda tryin' to dial it [being tough on the patients] down a little bit (Primary Care Provider, no other sample characteristics provided).[23]

Programme providers also recognised and reflected on the importance of establishing and maintaining good relationships and of giving positive reinforcement and encouragement and being supportive of weight-loss efforts.[20 23 30 36]

## Types of interaction/support offered

Several healthcare providers recognised that the intensity of interactions between programme staff and participants was important for motivating the latter to stay engaged and to sustain behaviour changes.[23 30] However, several provider participants raised concerns about the reality of this for their everyday clinical practice when time constraints were a real issue.[20 21 43] Other healthcare providers raised concerns around a lack of interdisciplinary working within clinic settings, which could inhibit their abilities to support weight loss, as well as lack of clarity with regard to professional role remits within teams:

> I work with our RN all the time so on a daily basis we talk about things going back and forth but the others [referring to dietitian and mental health workers] I don't really see to be honest (Nurse, no other sample characteristics provided).[21]

Although providers in the above study[21] raised broad issues in their interviews relating to these barriers, they reflected positively on the study WMP for facilitating interdisciplinary collaboration.

## Views about mode of support

When discussing preferred modes of support, healthcare providers considered issues regarding access and/or perceived effectiveness. Health providers in one primary care study[23] argued that telephone-delivered weight counselling was the most convenient for participants. In contrast, providers in another study (one that involved a residential WMP)[27] argued that face-to-face group interaction was essential and particularly useful for participants

with severe obesity who often experience social isolation. In another primary care study,[36] views regarding mode of delivery of support were more mixed. While recognising the practicalities of remote forms of support, programme providers (in this case nurses) argued that face-to-face interactions worked best for helping them connect more effectively and facilitated participant engagement and motivation. Some even stated that they did not regard remote support as support at all.

## Views about levels of provider engagement

Healthcare providers in one study[23] stated that they played a fairly peripheral role in aspects of programme delivery and that sometimes this made it difficult for them to fully engage with their patient and to assess progress. They suggested that individualised feedback from other professionals involved in programme delivery (eg, in this case weight-loss health coaches) would have been helpful. However, the study also reported that the majority of healthcare providers valued the fact that they played a limited role in the WMP, with time constraints and specific skill sets being raised as issues. Another study[36] raised related issues around level of provider engagement with aspects of the WMP. In this case, nurses discussed the perceived disadvantage of not being able to view the information provided to participants on the study website. Some stated that viewing this information would have allowed them to understand more fully, what participants were referring to in consultations. In one study,[43] GPs commented on and seemed to value the relatively 'loose' nature of the intervention design (in this case a weight management toolkit) as they considered it offered scope to enable them to tailor it to the individual and their community. Similarly, nurses in another study[36] expressed frustration around the lack of flexibility of their intervention, both in terms of how they were supposed to behave (ie, by not being directive) and also the lack of scope within the website to document individual issues. This was a concern raised by the participants themselves. Personnel in a residential WMP[27] specifically designed for people with severe obesity seemed to value having a very strict programme structure (in this case participants had to attend morning meetings, group activities and eat six meals a day at fixed times). The general feeling among staff was that instilling this strictness on participants would facilitate behaviours that they would then seek to maintain at home.

## Views about intervention content

While some (but not all), participants in one study[27] found personal development classes challenging and confrontational, providers in the same study consistently argued that personal development (ie, focussing on personal factors such as self-knowledge and self-acceptance) was essential and crucially important for maintaining lifestyle changes longer term:

> It is important that they become aware of what in their life makes a difference in being obese or

not (Personnel, no other sample characteristics provided).[27]

## DISCUSSION
### Principal findings
This review synthesised findings from qualitative data relating to the views of adults with BMI ≥35 kg/m$^2$ (and/or their healthcare providers) about engaging with WMPs. In summary, although there was variation expressed in views about the acceptability of various programme components (indicating the inappropriateness of a 'one size fits all' approach), there were, nevertheless, recurring themes around what both participant and programme providers described valuing and enjoying. Some of these key findings resonate with previous qualitative research with people with less severe obesity.[9 50]

Participants in our review described being attracted to WMPs that were perceived to be novel or exciting in some key way, as well as perceived to have been endorsed by their healthcare providers (a view supported by programme providers themselves). The sense of belonging to a group of people who shared similar issues relating to weight and food, and who had similar physiques and personalities, was described as being particularly important to many participants. This seemed to foster a strong group identity and related accountability, which seemed to help with motivation and continuing engagement.

Good relationships with programme providers were described as being highly valued, with ongoing encouragement and monitoring apparently important for facilitating motivation and behaviour change (a view also endorsed by the programme providers themselves). Group-based programme activities were enjoyed by many participants along with intensive support from programme providers. This observation is supported in previous qualitative research with people with less severe obesity.[9 50] However, in our review, concerns were raised about the availability of continuing support postintervention. Similarly, providers questioned the practicalities and logistics of integrating such intense support into their everyday clinical practices once the studies were completed.

Overall, both participants and programme providers valued having choice and flexibility. For example, participants welcomed flexibility around diet choices, flexibility around when face-to-face counselling sessions were scheduled, and welcomed personalised interventions. Similarly, some programme providers found the perceived lack of flexibility with various intervention components frustrating and prohibitive for supporting individualised care.

Those participants who described engaging in group discussions/therapy sessions and those who discussed engaging in exercises were mainly positive about their perceived benefits. Where it was discussed, participants valued the psychological input integrated into many interventions. This is a view supported in a study of user experiences of both Tier 2 and Tier 3 wt management services in England.[50] However, our review also highlighted that some participants did describe struggling with these aspects, with some describing them as particularly challenging. Some participants described difficulties with the various physical activities (because of a range of physical comorbidities). Not everyone enjoyed group interaction and discussions with others, sometimes apparently because they suffered from various mental health comorbidities.

### Practice implications
For intervention developers, it was clear from our review that social interaction activities tended to be valued. It was also apparent that ongoing encouragement and monitoring by programme providers was viewed as important for facilitating motivation and behaviour change. The waning intensity of programme activities and/or programme cessation could cause problems for maintaining behaviour change patterns if group interaction and support were integral components. There is a need for WMPs to help consumers to establish support postintervention.

Intervention developers should be aware that people with severe obesity might be especially vulnerable to both physical and mental comorbidities, which could inhibit engagement with certain intervention components (eg, group-based interaction; physical activities). This could inhibit their engagement with much fitter peers with fewer weight-related issues, or restrict their ability to undertake certain intervention components. This observation is less apparent in research with people with less severe obesity.[9] WMPs developers could consider including a choice of interaction styles/mix of physical activities to accommodate this.

### Strengths and limitations
To our knowledge, this is the first review of key findings from qualitative studies exploring participants' perspectives of WMPs for adults with severe obesity. Our review has highlighted a range of important factors that have the potential to facilitate engagement with WMPs for this group.

We were interested in ascertaining the views of participants with severe obesity (people with BMI≥35 kg/m$^2$). Therefore, our inclusion criteria were that papers needed to state that participants in their respective studies (ie, either in their qualitative evaluations or the intervention studies to which their qualitative evaluations were linked) had a mean BMI≥35 kg/m$^2$. Of those papers that only considered programme providers' views, these had to be linked to intervention studies where we could establish that included participants had a mean BMI≥35 kg/m$^2$. Only two papers stated that their respective WMPs were designed *specifically* for people with BMI≥35 kg/m$^2$.[24 42] Thus, across the papers, some people with BMI<35 kg/m$^2$ would have been included. Quotes from participants were not linked to specific detail regarding BMI status,

  

and so we cannot be certain that findings reflect exclusively the views of those with severe obesity.

Only nine papers linked participant quotes to sex;[24 27 29 31 35 36 38–40] only one to age status;[36] and none to socioeconomic/demographic characteristics, making it hard for us to consider whether any issues raised were particularly sensitive or pertinent to these aspects.

We know from a recent review of Tier 3 weight management interventions for adults with severe obesity that drop-out rates are very high (43%–63%).[51] Only four of our included papers stated that some of the participants in their qualitative evaluations had been low users, quitters or drop-outs[17 24 25 36] and only one of these papers linked quotes directly to intervention usage status.[36] Although our findings highlighted a range of views with regard to the usefulness or otherwise of various intervention components, it is worth noting that participant sample characteristics within the included papers are skewed towards those who had chosen to engage and who had completed the various intervention activities.

Applying quality criteria to qualitative research remains a contentious issue and there is no consensus regarding whether and how this should be done.[52 53] While authors of some qualitative evidence syntheses have chosen to exclude what they deem to be poor quality papers, we made the decision not to exclude any of the identified papers. We included 33 papers that each reported some qualitative data that met our inclusion criteria and addressed our key research questions. Although all included qualitative data, with regard to 'quality', some were deemed richer than others in terms of data and insights. Some ranged from being exclusively qualitative studies providing rich data in our areas of interest, through to studies that were actually primarily quantitative with responses to open-ended survey questions. The five studies providing qualitative data in the form of responses to open-ended survey questions within structured questionnaires[22 32 37 46 49] were deemed less useful as they presented only very limited qualitative data and insights. Despite this variation in the overall level of quality, we believed it was more important to retain any relevant findings rather than disregard based on study quality. In doing so, we would argue that all 33 papers contributed useful elements to the collective whole and enabled us to develop our understanding of the issues of importance to people with BMI ≥35 kg/m$^2$. We cannot exclude the possibility that unpublished service evaluations from within the NHS, that we failed to locate, might have been sources of rich data.

## Implications for research

No papers included in our review provided qualitative data from those who had been invited to join a WMP but who had declined to take part. Only four papers reported including participants who had not fully engaged with all programme activities to varying degrees. The views of those who do not engage are important and should be a focus of future research. In terms of pointers for effective interventions, it is worth acknowledging that key findings

will be skewed towards those who had chosen to engage and who had completed the various intervention activities. This review also demonstrated that the qualitative research literature focusing specifically on lifestyle WMPs for people with very high BMIs is limited, particularly for people who are low users or do not wish to engage with such services.

## CONCLUSIONS

WMPs that are perceived to be novel or exciting and WMPs that are perceived to be endorsed by healthcare providers tend to be valued by participants. The sense of belonging to a group of people who share similar issues and characteristics seems particularly important, helping to foster a strong group identity and related accountability—aiding motivation and continuing engagement. In-person group-based programme activities tend to be valued (over more remote forms of support), along with intensive support from programme providers. However, intervention developers should be aware that people with severe obesity might be especially vulnerable to both physical and mental comorbidities that could inhibit engagement with certain intervention components.

**Collaborators** The REBALANCE team: REBALANCE Project management team Elisabet Jacobsen (Health Economics Research Unit, University of Aberdeen, Aberdeen, UK), Dwayne Boyers (Health Services Research Unit, University of Aberdeen, Aberdeen, UK), David Cooper (Health Economics Research Unit, University of Aberdeen, Aberdeen, UK), Lise Retat (UK Health Forum, Fleetbank House, Salisbury Square, London, UK), Paul Aveyard (Nuffield Department of Primary Care Health Sciences, Oxford University, Oxford, UK), Fiona Stewart (Health Economics Research Unit, University of Aberdeen, Aberdeen, UK), Graeme MacLennan (Health Economics Research Unit, University of Aberdeen, Aberdeen, UK), Laura Webber (UK Health Forum, Fleetbank House, Salisbury Square, London, UK), Emily Corbould (UK Health Forum, Fleetbank House, Salisbury Square, London, UK), Benshuai Xu (UK Health Forum, Fleetbank House, Salisbury Square, London, UK), Abbygail Jaccard (UK Health Forum, Fleetbank House, Salisbury Square, London, UK), Bonnie Boyle (Health Economics Research Unit, University of Aberdeen, Aberdeen, UK), Eilidh Duncan (Health Economics Research Unit, University of Aberdeen, Aberdeen, UK), Michal Shimonovich (Health Economics Research Unit, University of Aberdeen, Aberdeen, UK), Cynthia Fraser (Health Economics Research Unit, University of Aberdeen, Aberdeen, UK), Lara Kemp (Health Economics Research Unit, University of Aberdeen, Aberdeen, UK). REBALANCE Advisory Group for all their advice and support during this project: Margaret Watson, Lorna Van Lierop, Richard Clarke, Jennifer Logue, Laura Stewart, Richard Welbourn, Jamie Blackshaw, Su Sethi.

**Contributors** AA and ZCS conceived the study idea for the qualitative systematic review. ZCS and MA-M screened all titles and abstracts. ZCS and MA-M conducted the data analysis and ZCS wrote the initial and subsequent manuscript drafts. AA, ZCS, MA-M, CR and MDB contributed critically to discussions about interpretation of data and revisions of manuscript drafts. AA, ZCS, MA-M, CR and MDB approved the final version.

**Funding** The project was funded by the NIHR Health Technology Assessment Programme (Project number: 15/09/04). See the HTA Programme website for further project information.

**Disclaimer** The views and opinions expressed therein are those of the authors and do not necessarily reflect those of the Department of Health, or the funders that provide institutional support for the authors of this report.

**Competing interests** None declared.

**Patient consent for publication** Not required.

**Provenance and peer review** Not commissioned; externally peer reviewed.

**Data availability statement** All data relevant to the study are included in the article or uploaded as supplementary information.

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
