## [Reviewer comments · BMJ Open]

ARTICLE DETAILS

TITLE (PROVISIONAL)	The acceptability and feasibility of weight management programmes for adults with severe obesity: A qualitative systematic review
AUTHORS	Skea, Zoë; Aceves-Martins, Magaly; Robertson, Clare; De Bruin, M; Avenell, Alison

VERSION 1 – REVIEW

REVIEWER	Katy Sutcliffe University College London, UK
REVIEW RETURNED	01-Mar-2019

GENERAL COMMENTS	Overview A really interesting qualitative evidence synthesis of the experiences of users and providers of weight-management services for people with severe obesity. The focus of this paper on severe obesity makes a valuable contribution to existing work in this area. One key value of the work is that it adds to the evidence-base by highlighting the similarities and differences between the population focus of this review (people with severe obesity) and the populations of other reviews (people with less-severe obesity). Another strength is the sensitive and thoughtful focus on the diversity of views, an explicit focus on negative cases which counter the prevalent viewpoint, for example about the value of social interaction. The focus on why particular elements of the programme are perceived to be important – i.e. the mechanisms through which these elements impact on decisions to join or stay in a WMP and how they impact on weight-loss behaviours – is also very strong. The writing is (largely speaking) exceptionally clear and accessible. I have given a few notes below on strategies to further strengthen the paper – the key one being to try to distil the findings a little further and to provide a summary account at the beginning of the findings section – ideally in a diagram – to answer their question ‘what is it about interventions that makes them helpful or unhelpful?’. Suggested amendments Abstract: The design section should refer to the synthesis and quality appraisal methods. This is important to inform readers – but also as methods for qualitative synthesis are still in development
---

	and it helps other researchers searching for qualitative evidence syntheses to identify reviews that use specific methods. Methods: Advisory group: Can you give us a bit more information about the REBALANCE advisory group? How were they identified? You mention that it 'included' lay members – but was it only lay people? Were there others? Who were the lay people – i.e. what experience / expertise did they bring? And what specific impacts did their advice have on the questions and areas of interest? Patient and public involvement is often not well reported in research – but the authors have a really valuable opportunity to showcase how important stakeholder engagement can be in determining the scope of a review. Excluded studies: Better signposting of the PRISMA diagram providing info on the flow of studies through the review is needed - unless the figure will be placed in the main body of the text? I had thought it was not there but only found it by chance at the end of the document when looking for something else. Findings:  - Overall the findings are well reported and illuminative but a little more distillation would help readability and hopefully achieve a greater depth of analysis. Providing readers early on in the paper with a summary of the themes that arose as particularly important – and/or some form of diagram / logic model depicting the key features and illustrating the mechanisms through which they are perceived to lead to successful weight management – would be enormously helpful. This work would be really useful for readers by providing a succinct summary of their narrative – but it would also enable the researchers to demonstrate how they have achieved conceptual / theoretical development with their work. See for example the diagram in - Archibald D, Douglas F, Hoddinott P, et al A qualitative evidence synthesis on the management of male obesity BMJ Open 2015. Cutting some of the words from the existing narrative to make space for this will sharpen the findings and enhance readability. - A second overarching issue is that the headings do not seem to illuminative / accurately reflect the text contained within them. Some consideration of how best to communicate your themes would be of value. - Structuring of themes is also an issue to consider. I appreciate it is always hard to work out how best to structure such complex findings, especially since you have both participant and provide viewpoints to consider. However, it felt slightly odd having the same issues come up
--	--

	time and again in three different places (findings participants, critical reflections participants, findings providers). One thing that would help readers is for the authors to be clear that this is the structure being used – and provide a justification for the approach – at the beginning of the findings section. An alternative would be (again with signposting about the structure at the beginning of the findings) to take each theme in turn and include participant views, critical reflections and provider views. This second option would have the benefit of a thorough investigation of each theme in turn – but with a lack of repetition / requirement of readers to remember what was said before about this particular theme. The clear demarcation of critical reflections is really valuable and should definitely remain in some form. Theme 1 – motivating factors: The narrative in this section is slightly confusing. You say that ‘Important ‘push’ factors were sometimes internal to participants’ – but then examples include the influence of experiences of other family members – which could be construed as external? You then move on to factors that were features of the programme. The descriptor ‘push factor’ makes more sense to me – and the programme features could be cast as ‘pull factors’? I.e. remove the confusing reference to ‘internal’? I may have missed something here though about the value of referring to ‘internal’ – if so greater clarity about this idea is needed. Theme 2 – WMP components:  - Subtheme ‘setting and context’ does not quite reflect what is in this section – i.e. how is the fact that a WMP is delivered one-to-one about context or setting? Could the heading be revised? - I’m not clear how the narrative under ‘type of interaction / support offered’ differs from the previous sub-theme of ‘importance of other people in the programme for accountability). Aside from the one point about flexible timing – the whole section ‘type of support offered’ (p.11, lines 19-35) seems to be about ‘importance of other people for accountability’ (or even ‘importance of social interaction for accountability’?) Perhaps the heading ‘type of support offered’ needs to reflect more clearly that it is getting at what underpins successful social interactions? And the narrative within the section needs to be more clearly tied to that idea.  - Also accountability seems to be a key concept in this section – and one that frequently comes up in views about facilitators of successful weight management – but a little more explanation of what is meant by this would be helpful for readers who have not come across the idea before.
--	---

	- Dietary components / programme tools - in efforts to reduce the length / tighten the narrative - sentences such as the following from the diet section could be removed - 'We examined data that were available from participants and/or programme staff relating to the perceived usefulness or otherwise of these dietary aspects' i.e. this is self-evident since you have provided a theme on this issue? Removing sentences like these will help to sharpen the findings. Discussion Implications for practice: this section makes no mention of encouraging social interaction – which seemed to be the strongest and most emphatically expressed theme from your work? Minor comments P3, line 4. Reference needed for the definition of severe obesity as being $\geq 35\text{kg/m}^2$. P6, lines 43-60. Could the paragraph at the bottom of page 6 be broken up a bit – it's one very long sentence which is hard to read especially with the citations punctuating the text. Also do we need all of the detail of different techniques – would a few examples suffice with the detail provided in the table of studies appendix? P6, line 50 – first instance of the phrase 'Behaviour Change Techniques' here needs to include the acronym in brackets – as the acronym is used several times at later points in the paper P8, lines 50-59. Another very long sentence (85 words!). This would be much more readable in smaller chunks.
--	---

REVIEWER	Elizabeth Sturgiss Monash University Melbourne Australia
REVIEW RETURNED	30-Apr-2019

GENERAL COMMENTS	Thank you for the opportunity to review this qualitative synthesis of papers reporting the acceptability and feasibility of weight management programs for people living with BMI over 35kg/m^2. It is a very well written paper and I was able to follow your study and the conclusions. As you have outlined there is a major limitation in this work - you are interested in people with BMI over 35kg/m^2, but the papers do not delineate the BMI of the participants and their quotes. This is a major difficulty with the method. You have concluded that the issues facing this population are similar to people with other BMIs, but it is difficult to come to this conclusion taking this limitation into account. I have been trying to think of ways around this in your method. One option would be to only include studies that had people over BMI 35 (there were two programs like this in your review), and compare the findings of these two to all of the others? Or contact the authors of papers for the BMI of particular quotes? This would be some work, but would add rigor to your findings. As it stands, I do not think you can conclude that your themes relate only to people with a BMI over 35kg/m^2.
---

	In addition this this feedback, I have some smaller suggestions:  - can you explain your coding process in more details? It is written that "papers" were grouped into themes but then quotes are presented in the paper. There needs to be sufficient detail for someone to replicate your method. - why have you chosen to focus on severe obesity? I agree with you that it is an important population, but it needs to be justified in your background - in your background "Effective weight loss services may reduce the need for bariatric surgery, and could also increase the effectiveness of subsequent bariatric surgery." I am not aware of any weight loss program that shows longterm weight loss. Please reference these statements. - what about non weight outcomes? Did this come up in your qualitative synthesis at all? It is a potential motivating factor for patients. - In the discussion "Good relationships with programme providers were described as being highly valued", I agree that this is an increasingly recognised factor. But I could not see it discussed in the results? Can you please expand this in the results? - in the methods you mention "Our pragmatic approach corresponded most closely to a 'realist' perspective" - the realist method is quite specific (see RAMASES group for examples and publications). I did not see your method and results cover realist in what would typically be expected. Thank you again for the opportunity to review your work. I hope the major limitation can be addressed as I agree that this is an important area of study.
--	--

REVIEWER	Catherine Spooner Centre for Primary Health Care & Equity, UNSW Sydney Australia
REVIEW RETURNED	30-Apr-2019

GENERAL COMMENTS	This was an interesting and well-written paper. My only comments relate to the discussion.  1. I think it would flow better if "Practice Implications" preceded "Strengths and Limitations" 2.. I do not know what "quotation data" is - can this be explained? 3. The paper would benefit from evidence-based suggestions for how to respond to the findings, rather than statements such as "intervention developers should bear in mind...". For example: Page 23, line 6: Perhaps there is a need for WMPs to help consumers to establish supports for after the WMP finishes Page 23, line 18: Perhaps WMPs could include physical activities that cater to all fitness/health levels 4. Implications for research: Problems with existing research are noted, but specific ideas for further research are not provided. Can
--

	these be provided? For example: research with people who have dropped out of WMPs to examine the reasons for drop out.
--	--

VERSION 1 – AUTHOR RESPONSE

Reviewer 1 comments	
A really interesting qualitative evidence synthesis of the experiences of users and providers of weight-management services for people with severe obesity. The focus of this paper on severe obesity makes a valuable contribution to existing work in this area.	Thank you for your positive comments.
Abstract: The design section should refer to the synthesis and quality appraisal methods. This is important to inform readers – but also as methods for qualitative synthesis are still in development and it helps other researchers searching for qualitative evidence syntheses to identify reviews that use specific methods.	Reference to the synthesis and the method of quality appraisal has now been added to the abstract.
Methods: Advisory group: Can you give us a bit more information about the REBALANCE advisory group? How were they identified? You mention that it 'included' lay members – but was it only lay people? Were there others? Who were the lay people – i.e. what experience / expertise did they bring? And what specific impacts did their advice have on the questions and areas of interest? Patient and public involvement is often not well reported in research – but the authors have a really valuable opportunity to showcase how important stakeholder engagement can be in determining the scope of a review.	We have added more detail regarding the advisory group to the methods section.
Excluded studies: Better signposting of the PRISMA diagram providing info on the flow of studies through the review is needed - unless the figure will be placed in the main body of the text? I had thought it was not there but only found it by chance at the end of the document when looking for something else.	We have now added better signposting to the PRISMA diagram at the start of the findings section.
Overall the findings are well reported and illuminative but a little more distillation would help readability and hopefully achieve a greater depth of analysis. Providing readers early on in the paper with a summary of the themes that arose as particularly important – and/or some form of diagram / logic model depicting the key	We have now incorporated a diagram (S1 Conceptual diagram) which attempts to illustrate the mechanisms that are likely to impact on decisions to join and decisions to stay in or drop out of WMPs.

features and illustrating the mechanisms through which they are perceived to lead to successful weight management – would be enormously helpful.	
A second overarching issue is that the headings do not seem to illuminative / accurately reflect the text contained within them. Some consideration of how best to communicate your themes would be of value.	Please see our responses to related comments below
Structuring of themes is also an issue to consider. I appreciate it is always hard to work out how best to structure such complex findings, especially since you have both participant and provide viewpoints to consider. However, it felt slightly odd having the same issues come up time and again in three different places (findings participants, critical reflections participants, findings providers). One thing that would help readers is for the authors to be clear that this is the structure being used – and provide a justification for the approach – at the beginning of the findings section.	We have now added an outline of the paper structure to the start of the findings section and have signposted readers to our illustrative diagram of key issues.
Theme 1 – motivating factors: The narrative in this section is slightly confusing. You say that ‘Important ‘push’ factors were sometimes internal to participants’ – but then examples include the influence of experiences of other family members – which could be construed as external? You then move on to factors that were features of the programme. The descriptor ‘push factor’ makes more sense to me – and the programme features could be cast as ‘pull factors’? I.e. remove the confusing reference to ‘internal’? I may have missed something here though about the value of referring to ‘internal’ – if so greater clarity about this idea is needed.	We agree that reference to the word ‘internal’ might be confusing to readers (for the reasons you outline) and so have re-worded as ‘personal.’
Theme 2 – WMP components: - Subtheme ‘setting and context’ does not quite reflect what is in this section – i.e. how is the fact that a WMP is delivered one-to-one about context or setting? Could the heading be revised?	We recognise that different review teams may interpret qualitative data in slightly different ways. We would like to keep reference to ‘setting’ as we were referring to comments participants made about liking where the WMP was set (e.g. a football club). However, for clarity we have replaced ‘context’ with ‘style’ in an attempt to better reflect the content of this section – e.g. we were referring here to comments participants made about e.g. liking that their WMP did not seem to be overly focussed on dieting etc

I'm not clear how the narrative under 'type of interaction / support offered' differs from the previous sub-theme of 'importance of other people in the programme for accountability). Aside from the one point about flexible timing – the whole section 'type of support offered' (p.11, lines 19-35) seems to be about 'importance of other people for accountability' (or even 'importance of social interaction for accountability'?) Perhaps the heading 'type of support offered' needs to reflect more clearly that it is getting at what underpins successful social interactions? And the narrative within the section needs to be more clearly tied to that idea.	In the type of interaction/support offered section we were referring mainly to comments made about valuing fairly intense support from staff; valuing face-to-face over more remote forms; tending to valuing group based activities over one-to-one activities. Whilst we agree that this is related to the previous section 'importance of other people in the programme for accountability' it differs in the sense that the previous section covers comments about valuing being part of a similar group of people (who share similar issues/problems). This was important for fostering strong group identities/feelings of accountability – as such, we would like to leave the headings as they are.
Also accountability seems to be a key concept in this section – and one that frequently comes up in views about facilitators of successful weight management – but a little more explanation of what is meant by this would be helpful for readers who have not come across the idea before.	We did find this to be a key concept. People described feeling accountable or responsible to other participants and programme providers – This was something that was apparently important for people in terms of motivating them to stick with the programmes and to not let their fellow participants down by dropping out or not sustaining behaviour changes. We have added clarity to this section.
Dietary components / programme tools - in efforts to reduce the length / tighten the narrative - sentences such as the following from the diet section could be removed - 'We examined data that were available from participants and/or programme staff relating to the perceived usefulness or otherwise of these dietary aspects' i.e. this is self-evident since you have provided a theme on this issue? Removing sentences like these will help to sharpen the findings.	We have removed this sentence and a similar sentence from the start of section f).
Discussion Implications for practice: this section makes no mention of encouraging social interaction – which seemed to be the strongest and most emphatically expressed theme from your work?	We have now added an opening sentence in this section to reflect this.
Minor comments P3, line 4. Reference needed for the definition of severe obesity as being $\geq 35\text{kg/m}^2$. P6, lines 43-60. Could the paragraph at the bottom of page 6 be broken up a bit – it's one	There is no official term for $\text{BMI} \geq 35\text{kg/m}^2$. Obesity starts at $\text{BMI} \geq 30\text{kg/m}^2$ and 'morbid' obesity $\geq 40\text{kg/m}^2$ (the term 'morbid' is rightly falling out of use). We used the term 'severe' obesity to differentiate it from 'morbid' obesity. $\geq 35\text{kg/m}^2$ is often the lower bound of the cut-off for clinical weight management services or

very long sentence which is hard to read especially with the citations punctuating the text. P6, line 50 – first instance of the phrase 'Behaviour Change Techniques' here needs to include the acronym in brackets – as the acronym is used several times at later points in the paper P8, lines 50-59. Another very long sentence (85 words!). This would be much more readable in smaller chunks.	bariatric surgery. We have made it clear in the paper that this is our terminology. This paragraph has now been broken up. This has now been added. This sentence has now been broken up.
Reviewer 2	
Thank you for the opportunity to review this qualitative synthesis of papers reporting the acceptability and feasibility of weight management programs for people living with BMI over 35kg/m². It is a very well written paper and I was able to follow your study and the conclusions. As you have outlined there is a major limitation in this work - you are interested in people with BMI over 35kg/m², but the papers do not delineate the BMI of the participants and their quotes. This is a major difficulty with the method. You have concluded that the issues facing this population are similar to people with other BMIs, but it is difficult to come to this conclusion taking this limitation into account. I have been trying to think of ways around this in your method. One option would be to only include studies that had people over BMI 35 (there were two programs like this in your review), and compare the findings of these two to all of the others? Or contact the authors of papers for the BMI of particular quotes? This would be some work, but would add rigor to your findings. As it stands, I do not think you can conclude that your themes relate only to people with a BMI over 35kg/m².	Many thanks for your positive comments. Although we state in the discussion that “Quotes from participants were not linked to specific detail regarding BMI status, and so we cannot be certain that findings reflect exclusively the views of those with severe obesity” we can be confident that our findings come from studies where the majority of participants had severe obesity. To clarify our inclusion criteria: papers needed to state that participants in their respective studies (i.e. either in their qualitative evaluations or the intervention studies to which their qualitative evaluations were linked) had a mean BMI $\geq 35\text{kg/m}^2$. We did not include any papers that stated the mean BMI was less than this. In addition, BMI for those involved in qualitative evaluations was reported in nine papers. Of those that provided a mean, this ranged from 36.8-44.7kg/m². So although individual quotes did not state BMI we can still be confident that our findings reflect the views of those with severe obesity.
In addition to this feedback, I have some smaller suggestions: - can you explain your coding process in more details? It is written that "papers" were grouped	For qualitative data analysis and write up, it is fairly standard to discuss key themes emerging within and across papers and then to summarise the key themes in a paper along with

into themes but then quotes are presented in the paper. There needs to be sufficient detail for someone to replicate your method.	illustrative quotes. For clarity, we have added more detail to the methods section.
- why have you chosen to focus on severe obesity? I agree with you that it is an important population, but it needs to be justified in your background	Further justification for our focus on severe obesity has now been provided in the background section.
- in your background "Effective weight loss services may reduce the need for bariatric surgery, and could also increase the effectiveness of subsequent bariatric surgery." I am not aware of any weight loss program that shows longterm weight loss. Please reference these statements.	A reference for these statements has now been provided.
what about non weight outcomes? Did this come up in your qualitative synthesis at all? It is a potential motivating factor for patients.	For this particular review, we did not focus on weight outcomes (i.e. we were not focussed on whether the WMPs resulted in weight loss) but rather we were interested in the factors that people described valuing during their engagement with the various WMPs. In terms of non-weight outcomes, for example, across several papers people did discuss the positive psychological benefits they experienced by taking part and we discuss this in section e) p. 12
- In the discussion "Good relationships with programme providers were described as being highly valued", I agree that this is an increasingly recognised factor. But I could not see it discussed in the results? Can you please expand this in the results?	14 papers did discuss this. In particular, participants seemed to value the positive, friendly, and non-judgemental encouragement they received from providers. We discuss this on p.10 and provide 2 illustrative quotes.
- in the methods you mention "Our pragmatic approach corresponded most closely to a 'realist' perspective" - the realist method is quite specific (see RAMASES group for examples and publications). I did not see your method and results cover realist in what would typically be expected.	We did not want to claim that we had conducted a realist synthesis but rather that we drew on a realist perspective (we have edited the wording in the analysis section to reflect this). At the same time, our approach was informed by and used aspects of review methods such as thematic synthesis and analytical approaches developed from methods of inquiry such as grounded theory. We have added more detail to the analysis section to explain our approach.
Reviewer 3	
My only comments relate to the discussion. 1. I think it would flow better if "Practice Implications"preceded "Strengths and Limitations"	This section now precedes the 'Strengths and limitations section.

2. I do not know what "quotation data" is - can this be explained?	We agree this might cause confusion and so have removed and replaced with 'quotes'
3. The paper would benefit from evidence-based suggestions for how to respond to the findings, rather than statements such as "intervention developers should bear in mind...". For example: Page 23, line 6: Perhaps there is a need for WMPs to help consumers to establish supports for after the WMP finishes Page 23, line 18: Perhaps WMPs could include physical activities that cater to all fitness/health levels	Thank you – we agree with these helpful suggestions. We were subtly making the points you raise, but agree that additional pointers would be helpful – these have now been added.
4. Implications for research: Problems with existing research are noted, but specific ideas for further research are not provided. Can these be provided? For example: research with people who have dropped out of WMPs to examine the reasons for drop out.	We agree this is important and have added an additional sentence to this section.

VERSION 2 – REVIEW

REVIEWER	Katy Sutcliffe UCL Institute of Education, UK
REVIEW RETURNED	24-Jul-2019

GENERAL COMMENTS	The authors have satisfactorily addressed the vast majority of points that I previously made. However, a couple of points that would be simple to address remain outstanding. Once these are addressed I feel the paper is ready for publication. Describing accountability – in the response to reviewer comments the authors make clear that accountability entails 'feeling accountable or responsible to other participants and programme providers'. However, the revisions to the paper itself do not contain the important qualifier 'to other participants and programme providers'. Please could this be added to page 10 lines 43-44 it would make the description much clearer. P3, line 4. Reference needed for the definition of severe obesity as being $\geq 35\text{kg/m}^2$ – in the response to reviewer comments the authors state that they make it clear that this is their terminology, however I still feel a reference for defining severe obesity is critical. What is their justification / authority for suggesting $\geq 35\text{kg/m}^2$ equates to severe obesity? Is this definition consistent with others understanding of severe obesity? Would this reference suffice? - https://onlinelibrary.wiley.com/doi/epdf/10.1111/obr.12601. Regardless, a reference to support the claim that levels of severe obesity are increasing is most definitely needed. In addition, there is a typo in the edits made as the brackets are not closed – i.e.
---

	'There has been a continued increase in body mass index $\geq 35\text{kg/m}^2$ (which we call here 'severe obesity' in adults worldwide).'
--	--

REVIEWER	Catherine Spooner UNSW Australia
REVIEW RETURNED	22-Jul-2019

GENERAL COMMENTS	This paper provides an interesting synthesis of qualitative data relating to the experience of participating in weight management programs for adults with a body mass index $\geq 35\text{kg/m}^2$ and for the providers of these programs. While the authors have responded in some way to the reviewers' comments, the paper still needs significant editing to warrant publication in this journal. Specific areas to be addressed are suggested below. Editing is required to improve grammatic, readability and academic style. For example:  1. There are numerous overly long sentences, sometimes constituting a whole paragraph. E.g.: 'With the exception of one study, in which some GPs (but not all) were reportedly less enthusiastic, {24} views about being involved in a WMP were generally very positive, with health professionals acknowledging that engagement was potentially very useful for them in terms of facilitating a conversation around weight loss with participants, and recognising that this can often be challenging in their everyday practices. 2. Hyphens could be used often to add clarity with compound adjectives, e.g.: in-person group-based activities; residential weight-loss centre 3. the term 'felt' is not appropriate unless discussing feelings. 4. There is overuse of the word 'also', which can generally be deleted. 5. There is overuse of the expression 'In terms of', which can generally be deleted. 6. Tense needs to be consistently past tense. 7. The term 'fairly intensive support' is used a number of times – what does this mean? 8. Some overly enthusiastic language is used e.g.: participants very much valued the psychological input integrated into many interventions; 'A strong recurring theme' 9. The terms 'clearly' or 'it is clear that' should generally not be used – what is clear to one person is not clear to another and it can generally be deleted. 10. Expressions such as 'it is worth noting that" and 'this is worthy of note' are neither useful nor appropriate. 11. 'In terms of' is an expression that has been overused and generally can be deleted. 12. Other terms that are inappropriate are 'unsurprisingly' 'Evidence synthesis' has been replaced by 'systematic review' in the heading but 'evidence synthesis' continues to be used throughout the document. Consistent terms should be used. Re the statement: 'public health guidance excludes evidence on weight loss programmes for obese people with co-morbidities'; is this in the UK or worldwide? In the description of studies, theories are listed as techniques. Theories are not techniques: determination theory based support regulatory focus theory ... social learning theories. One quote includes the term 'oof' - can the meaning of this term be added?
--

	This sentence is unclear and needs to be rewritten: ‘Although views were sometimes mixed, participants tended to describe valuing the flexibility and variety of diet format.’ The quote beginning “I think [having a set meal plan to follow]” ...” was described as being a quote that ‘illustrates that participants often discussed appreciating when programmes apparently emphasised changing attitudes towards food and eating over promoting a specific diet per se.’ I do not think the quote does illustrate anything about the frequency of participants raising this issue. Are you saying this quote reflects something that was reflected in multiple studies? The discussion repeatedly talks about what ‘perhaps’ might be needed when there is extensive literature on behaviour change and weight management that can be drawn upon to substantiate the recommendation. E.g. Perhaps there is a need for WMPs to help consumers to establish supports post intervention. The discussion of modes of support conflates comments about access with comments about effectiveness. The modes discussed reportedly have benefits for one, perhaps at the expense of the other. This section needs to be rewritten. It is already two years since the end date of the studies included in the review. If this is to be amended and published, I suggest this be done very quickly before it is out of date.
--	--

VERSION 2 – AUTHOR RESPONSE

Reviewer 1 comments	
There are numerous overly long sentences, sometimes constituting a whole paragraph. E.g.: ‘With the exception of one study, in which some GPs (but not all) were reportedly less enthusiastic, {24} views about being involved in a WMP were generally very positive, with health professionals acknowledging that engagement was potentially very useful for them in terms of facilitating a conversation around weight loss with participants, and recognising that this can often be challenging in their everyday practices.	We have edited this sentence and numerous other sentences throughout the manuscript in an attempt to shorten overly long sentences.
Hyphens could be used often to add clarity with compound adjectives, e.g.: in-person group-based activities; residential weight-loss centre	These have been added throughout the manuscript.
the term ‘felt’ is not appropriate unless discussing feelings.	The term ‘felt’ has been removed.

There is overuse of the word 'also', which can generally be deleted.	The word 'also' has been removed in several places throughout the manuscript.
There is overuse of the expression 'In terms of', which can generally be deleted.	'In terms of' has been removed in several places throughout the manuscript.
Tense needs to be consistently past tense	This has been amended.
The term 'fairly intensive support' is used a number of times – what does this mean?	To avoid confusion the word 'fairly' has now been removed.
Some overly enthusiastic language is used e.g.: participants very much valued the psychological input integrated into many interventions; 'A strong recurring theme'	We have edited these sentences in an attempt to soften the language.
The terms 'clearly' or 'it is clear that' should generally not be used – what is clear to one person is not clear to another and it can generally be deleted.	These terms have now been removed in various places throughout the manuscript.
Expressions such as 'it is worth noting that' and 'this is worthy of note' are neither useful nor appropriate.	These expressions have been removed.
'In terms of' is an expression that has been overused and generally can be deleted.	This expression has been removed in several places.
Other terms that are inappropriate are 'unsurprisingly'	This term has been removed.
'Evidence synthesis' has been replaced by 'systematic review' in the heading but 'evidence synthesis' continues to be used throughout the document. Consistent terms should be used.	The manuscript has now been edited for consistency.
Re the statement: 'public health guidance excludes evidence on weight loss programmes for obese people with co-morbidities'; is this in the UK or worldwide?	This is referring to the UK. We have edited the manuscript to reflect this.
In the description of studies, theories are listed as techniques. Theories are not techniques: determination theory based support regulatory focus theory ... social learning theories.	The manuscript has now been edited to reflect this.

One quote includes the term 'oof' - can the meaning of this term be added?	This was a verbatim quote provided by the study authors and they did not provide clarity on the exact meaning of the expression 'oof'. However, within the context of the quote we interpreted it as meaning that a WMP being endorsed by and delivered at the surgery gave this person a boost. We have added comment of this after the specific quote.
This sentence is unclear and needs to be rewritten: 'Although views were sometimes mixed, participants tended to describe valuing the flexibility and variety of diet format.'	We have edited this sentence for clarity.
The quote beginning "I think [having a set meal plan to follow]"..." was described as being a quote that 'illustrates that participants often discussed appreciating when programmes apparently emphasised changing attitudes towards food and eating over promoting a specific diet per se.' I do not think the quote does illustrate anything about the frequency of participants raising this issue. Are you saying this quote reflects something that was reflected in multiple studies?	In this quote, the participant was making the point that in the POWeR+ WMP the emphasis seemed to be on trying to change attitudes and thinking over promoting a set diet. This was a comment that was reflected in other studies. We have re-organised this section and added the relevant references for clarity.
The discussion repeatedly talks about what 'perhaps' might be needed when there is extensive literature on behaviour change and weight management that can be drawn upon to substantiate the recommendation. E.g. Perhaps there is a need for WMPs to help consumers to establish supports post intervention.	We have removed the words 'perhaps' in an attempt to strengthen our recommendations.
The discussion of modes of support conflates comments about access with comments about effectiveness. The modes discussed reportedly have benefits for one, perhaps at the expense of the other. This section needs to be rewritten.	When discussing preferred modes of support, health care providers (across 3 studies) did indeed consider issues regarding access and/or perceived effectiveness. We have added a sentence to the start of the relevant paragraph on p.19.
Reviewer 2	
Describing accountability – in the response to reviewer comments the authors make clear that accountability entails 'feeling accountable or responsible to other participants and programme providers'. However, the revisions to the paper itself do not contain the important qualifier 'to other participants and programme providers'.	We have now added this qualifier.

Please could this be added to page 10 lines 43-44 it would make the description much clearer.	
Reference needed for the definition of severe obesity as being $\geq 35\text{kg/m}^2$ – in the response to reviewer comments the authors state that they make it clear that this is their terminology, however I still feel a reference for defining severe obesity is critical. What is their justification / authority for suggesting $\geq 35\text{kg/m}^2$ equates to severe obesity? Is this definition consistent with others understanding of severe obesity? Would this reference suffice? - https://onlinelibrary.wiley.com/doi/epdf/10.1111/obr.12601. Regardless, a reference to support the claim that levels of severe obesity are increasing is most definitely needed. In addition, there is a typo in the edits made as the brackets are not closed – i.e. ‘There has been a continued increase in body mass index $\geq 35\text{kg/m}^2$ (which we call here ‘severe obesity’ in adults worldwide.’	We have now added 2 new supporting references to the introduction.